# Advances in Hazelnut (*Corylus avellana* L.) Rootstocks Worldwide

**Mercè Rovira**

Institut de Recerca i Tecnologia Agroalimentàries (IRTA) Fruit Production, Mas Bové, Ctra. Reus-El Morell Km 3,8, 43120 Constantí, Spain; merce.rovira@irta.cat

**Abstract:** Studies on hazelnut (*Corylus avellana* L.) rootstocks have been limited to date. However, the use of vigorous, non-suckering rootstocks for this species could increase the cost-effectiveness of orchards by reducing the annual need to prune suckers, thus facilitating mechanical harvesting, and reducing orchard management costs and environmental impact. Seedlings of the non-suckering Turkish tree hazel (*C. colurna* L.) have been used traditionally in Serbia. In the 1970s, the United States Department of Agriculture in Corvallis, Oregon (USA) released the first two non-suckering clonal rootstocks—'Dundee' and 'Newberg'—from open-pollinated seeds of *C. colurna*. Moreover, selection of *C. avellana* cvs. with few suckers is continuing. Trials carried out in different countries with own-rooted and grafted plants have shown good performance of grafted hazelnuts. Currently, some nurseries in several countries are propagating hazelnut rootstocks and grafting trees for planting commercial orchards. Interest in these cultivar/rootstock combinations is increasing, and more new orchards of grafted trees are expected to be planted in the coming years.

**Keywords:** hazelnut; rootstocks; general overview





## 1. Introduction

The natural habit of the European hazelnut (*Corylus avellana* L.) is a large multi-stemmed shrub, which annually produces suckers from buds located at the base of the trunk during the growing season. To facilitate management and mechanized harvest in commercial orchards, growers in several producing countries (Oregon (USA), France, Chile, and to a lesser extent, Spain and Italy) train hazelnuts to a single trunk. This training system makes it necessary to eliminate the suckers that continually appear each growing season. In large orchards, sucker removal becomes a major cultural operation, requiring four to five herbicide sprays per year and occasional hand removal in winter. This represents a significant expenditure of time and money [1–5]. The progressive reduction over the years in the number of herbicides that are available, effective, and have less environmental impact has made manual removal of suckers still common and thus costly. This situation could be improved by using non-suckering rootstocks in commercial hazelnut orchards. The authors of [5] considered suckering habit as one of the problems of hazelnut growing, and proposed, for Mediterranean countries, the selection and use of *C. colurna* L. as a non-suckering rootstock, a proven approach in the Balkan countries. The same idea was expressed by [6] as the need for the Spanish and Italian hazelnut industry to move from multi-trunk trees to the single-trunk trees using *C. colurna* rootstocks. More recently, [7] proposed the use of non-suckering rootstocks, together with other orchard management practices (subirrigation systems and mechanical pruning), as new technologies to be adopted by hazelnut producers. In addition, [8] considered non-suckering rootstocks to be very beneficial for commercial production. Despite the interest shown by the hazelnut community in the use of non-suckering rootstocks, studies are limited in the genus *Corylus*. Some research was begun in the middle of the 20th century, and some continues today.

The species *C. colurna*, the best known of the tree hazels, is native to the Balkan Peninsula, Turkey, the Caucasus, and northwestern Iran [8]. As a specimen plant, it

is usually a tall, pyramidal tree displaying winter catkins and an attractive scaly bark. Experience with *C. colurna* in the USA has shown that forms of this species are more drought tolerant and cold hardy than *C. avellana* cultivars. Its non-suckering characteristic gives *C. colurna* potential as a rootstock. In addition, sucker control could be obtained using other non-suckering *Corylus* species (*C. chinensis* Franch, *C. jacquemontii* Decne, *C. fargesii* (Franch.) C. K. Schneid) or interspecific hybrids as seedlings or clones [1,9]. Until now, three different types of hazelnut rootstocks have been investigated: seeds from selected *C. colurna* trees, two clonal selections ('Dundee' and 'Newberg') from open pollination of *C. colurna*, and selected *C. avellana* cultivars that are vigorous and produce few suckers.

This review presents information on the different *Corylus* materials used as rootstocks for *C. avellana*, the results of rootstock trials carried out in several countries, and the different methods of grafting hazelnuts. Finally, a general overview of future trends related to the use of rootstocks in commercial hazelnut orchards is presented.

## 2. Hazelnut Rootstock Materials

### 2.1. Corylus colurna L. Seedlings as Rootstocks

The main experience with seedlings of this species comes from the United States Department of Agriculture in Corvallis, Oregon, where research over several decades (1940–1970) has demonstrated the advantages and disadvantages of *C. colurna* seedlings as rootstocks (Table 1). The advantages are the non-suckering habit and the deep root system. Moreover, the graft unions produce abundant callus and heal readily. Seedlings of *C. colurna* are graft-compatible with all cultivars and *Corylus* species tried. Due to differences in bark color and texture, the union between the *C. colurna* rootstock and *C. avellana* scion is readily evident [1]. However, as a nursery tree, *C. colurna* has several disadvantages. Its seeds may take two or more years to germinate. After germination, the seedling trees often require two additional years of growth before reaching a size suitable for grafting. The tree produces a strong tap root with a few stiff, lateral roots. This makes digging, handling, shipping, and transplanting difficult. This poor root system could adversely influence transplant survival, as well as lengthen the time for the tree to become established in the orchard and begin growth. In addition, the Turkish tree hazel is difficult to propagate asexually. Propagation by cuttings has been unsuccessful, and by layering is difficult. These factors reduce its chances for routine use as a clonal or seedling rootstock by commercial nurseries. In Oregon, trees in old orchards established on *C. colurna* rootstocks many years ago are frequently more variable in size and yield than self-rooted trees of *C. avellana* [10]. Thus, pure *C. colurna* seedlings are no longer considered suitable as rootstocks for commercial orchards in the USA because of difficulties in the nursery and adverse elects on yield of some cultivars [9]. Nevertheless, *C. colurna* seedlings continue to be used as rootstocks for the ornamentals *C. avellana* var. *contorta* Bean and *C. avellana* var. *pendula* H.Jaeger, where yield reduction is of no consequence.

**Table 1.** Advantages and disadvantages of *Corylus colurna* L. seedlings as a rootstock [1].

| Advantages | Disadvantages |
| --- | --- |
| Non-suckering habit | Seeds need two or more years to germinate |
| Deeply rooted | Seedlings need to grow two or more years before attaining graftable size |
| Abundant callus after grafting | Tree with a strong tap root with a few stiff lateral roots |
| Graft-compatible with *C. avellana* cultivars | Poor root system |
| Differences in bark color and texture with *C. avellana* | |

Despite the above-mentioned disadvantages of *C. colurna* seedlings as rootstocks, they have been used in Serbia since 1972, and evaluation of genetic diversity in the species started in 1983 at the Faculty of Agriculture in Novi Sad [11]. Within the species, individual

trees have been selected that give good seed germination, non-suckering and high-quality seedlings, and compatibility as rootstocks with well-known *C. avellana* cultivars [12,13]. Using English grafting, different hazelnut cvs. were grafted and good results were obtained. Two- or three-year-old seedlings are recommended for use in grafting [4], although among the different forms of *C. colurna* evaluated, some seedlings appear to be suitable for grafting already at the end of the first year [14,15]. Seedlings of selected *C. colurna* trees (NS A2, NS B4) are high-quality rootstocks [4] and the grafted trees are long-lived, more vigorous, and more productive than self-rooted trees, are resistant to frost and drought, and are adapted to a wide range of soil conditions [15–17].

Seedlings of *C. colurna* were evaluated with four self-rooted cultivars ('Cosford', 'Istarski Dugi', 'Rimski' and 'Tonda Gentile Romana') in a region of Eastern Serbia characterized by high air temperatures (30–32 °C), frequent droughts, and fairy low rainfall. The results showed larger nuts and higher nut weight in trees grafted on *C. colurna* seedlings and the trees were more vigorous and productive [16].

In 2013, plants of 'Tonda Gentile delle Langhe' and the pollinizer 'Red Lambert' grafted in a nursery in Sabac (Novi Sad, Serbia) on *C. colurna* seedlings were planted in an experimental field in Monferrato and Langhe (Piedmont, Italy) and early results showed good adaptation of the plants [18]. Additionally, in Italy, at the University of Perugia, a trial was set up in 2014 to compare self-rooted trees of Italian cvs. ('Tonda Gentille delle Langhe', 'Tonda di Giffoni', 'Tonda Romana' and 'Tonda Francescana') with trees grafted to *C. colurna* seedling rootstocks [19]. The preliminary results indicate that the grafted trees have a smaller canopy, more upright growth, and compact bearing. In Piacenza (Italy), preliminary data from a trial carried out in 2017 in the experimental greenhouse of the Università Cattolica del Sacro Cuore with cvs. 'Tonda Gentile delle Langhe', 'Tonda di Giffoni', and 'Tonda Romana', own-rooted and grafted plants on *C. colurna* seedlings, planted in pots, suggest a minor competition between reproductive and vegetative activity in the grafted plants [20]. In that trial, trees grafted to *C. colurna* seedlings had lower shoot lengths and bore a high density of female and male flowers in comparison with sucker-propagated plants. Furthermore, to better exploit the particularities of *C. colurna* seedlings as rootstocks, research projects are being developed in Italy to assess the adaptability to soils with a high content of active limestone, increase the density of plants per ha, and improve water use efficiency [21].

### 2.2. Clonal Selections from Open-Pollinated Corylus colurna L.

Rootstock breeding started in Oregon in 1968. In nursery rows of open-pollinated *C. colurna* seedlings, those whose traits appeared to be intermediate between *C. colurna* and *C. avellana* were selected and repropagated. Over 20 years, about 150 potential interspecific hybrids were selected from among 20,000 seedlings raised. Two selections were released as clonal rootstocks and given names: 'Newberg' (USOR 7-71) and 'Dundee' (USOR 15-71) [22]. 'Dundee' has light-colored, silvery bark that is quite smooth, while the bark of 'Newberg' is also light but is rougher, intermediate between that of 'Dundee' and *C. colurna*. Both impart vigor to the scion cultivar and are non-suckering. Both rootstocks are considered interspecific hybrids because their nut and husk characteristics differ from those of the maternal parent. Despite their promise, these clonal rootstocks have been little used in the main producing countries. Hazelnut growers in Oregon (USA), where the first selection work for non-suckering rootstocks was conducted, have shown little interest in this innovation. Unfortunately, 'Dundee' and 'Newberg' are highly susceptible to eastern filbert blight (EFB) caused by *Anisogramma anomala* (Peck) E. Müll. [8], the main fungal disease in the USA [9]. Although these two rootstocks produce few suckers, there is a disadvantage to propagating any scion on a rootstock that suckers, since the suckers could be sources of infection by the fungus [23]. As a result, the nursery industry in USA has continued to produce self-rooted trees by layerage, and since 2005, by micropropagation. Other research has been conducted at the University of Torino (Italy), where eight materials

were selected as non-suckering rootstocks from seedlings obtained by crossing *C. colurna* x *C. avellana* [24,25].

In 2000, a rootstock trial was established at IRTA-Mas Bové, Constantí, Tarragona (northeastern Spain), using the main hazelnut cultivar in that area 'Negret' (low vigor, late to come into bearing, sensitive to iron chlorosis, and high sucker emission), grafted on four different rootstocks, including 'Dundee' and 'Newberg', compared to self-rooted 'Negret' as the control. 'Dundee' and 'Newberg' rootstocks improved the agronomic performance of 'Negret', reducing the number of suckers while increasing productivity and vigor, resulting in higher nut yields at lower costs [26,27]. In addition, the trial showed that grafted trees are more resistant to iron chlorosis and retain their leaves for a longer time, an important aspect in that the grafted trees can take up soil nutrients for a longer period of time.

'Dundee' and 'Newberg' were also imported into Australia in 2008 for grafting [28], but results have not yet been reported.

Lately, different research centers and universities around the world are conducting studies on hazelnut rootstocks. In Italy, two universities recently started some trials. At the University of Torino, a trial of 'Tonda Gentile delle Langhe' grafted on 'Dundee' and 'Newberg' was established in 2017 to compare grafted trees, self-rooted single stem trees (monocaule), and self-rooted multi-stem bushes (policaule) (Valentini pers. comm). In 2016, the University of Tuscia planted some 'Dundee' plants in the hazelnut collection of "Le Cese" and has ongoing research on different Italian cvs. grafted on 'Dundee' rootstocks, with the aim to facilitate sucker control for more sustainable hazelnut cultivation (Silvestri, pers. comm). In 2016–2017, the University of Craiova in Valcea (Romania) grafted three cultivars from their breeding program ('Cozia', 'Uriase de Valcea', 'Valcea 22') and other cultivars ('Hall's Giant', 'Romavel', and 'Tonda Gentile delle Langhe') to 'Dundee' and 'Newberg' rootstocks. The studies are ongoing (Botu, pers. comm).

### 2.3. Corylus avellana L. Rootstocks

Some research has been carried out using different vigorous *C. avellana* cultivars that produce few suckers. In the USA, [1] established a 'Daviana' rootstock trial with the scion cvs. 'Butler', 'Ennis', 'Jemtegaard 5', 'Lansing', and 'Ryan', with 'Barcelona' as the check. Moreover, [29] suggested the use of vigorous seedlings of 'Merveille de Bolwiler' with few suckers. One of the first trials of hazelnut rootstocks in Europe was established in Nebrosi, Sicilia (Italy) in 1970, to compare self-rooted and grafted plants. Four Italian cvs. ('Carrello', 'Santa Maria del Gesù', 'Tonda Gentile delle Langhe' and 'Tonda Romana') were grafted on suckers of the Sicilian cv. 'Santa Maria del Gesù'. After 12 years, the self-rooted trees showed better vegetative and productive behavior than the grafted trees [30]. At IRTA-Mas Bové (northeastern Spain), a trial was planted in 1989 with 'Negret' as the scion cultivar grafted to seedling rootstocks, and with self-rooted 'Negret' as the control. Seedlings of seven *C. avellana* cultivars ('Daviana', 'Gironell', 'Grifoll', 'Grifoll Fatarella', 'Merveille de Bolwiler', 'Queixal de gos', and 'Tonda Bianca') were included in the trial. The outstanding result for vegetative growth and agronomic behavior was for 'Negret' grafted to seedlings of 'Tonda Bianca', a cultivar from Campania (Italy). From among these seedlings, clonal rootstock 'TB-69' was selected [31]. In Karaj (Iran), some local genotypes of *C. avellana* suitable as rootstocks and tolerant to drought and low humidity have been selected [32].

Moreover, in the hazelnut collection at "Le Cese" surrounding the basin of Lago di Vico in Viterbo province (Italy), [33] noted some cultivars with a low emission of suckers: 'Closca Molla', 'Heynick's Zellernuss' and 'Pallagrossa'. They suggested their use as non-suckering rootstocks for other hazelnut cultivars. At the University of Torino, two seedlings from crosses of 'Tonda Gentille delle Langhe' as the female parent with two different pollen parents were selected for production of no suckers [25].

In 2010, researchers at the Instituto de Investigaciones Agropecuarias (INIA) in Temuco (Araucanía region, southern Chile) planted an experimental orchard with own-rooted and grafted trees. Their main cv. 'Tonda di Giffoni' was the scion and 'Daviana' was the pollinizer [34]. These materials were grafted to the rootstock BA-5, a clone of 'Chilean

Barcelona'. The aim of the trial was to reduce the unproductive (juvenile) time and improve the production of pollen and nuts. The preliminary results of the study indicated that grafting technique can shorten the unproductive period of 'Tonda di Giffoni' and increase the production of catkins and pollen, compared to the own-rooted trees. The trials mentioned above and carried out at different universities and research centers in several countries are summarized in Table 2.

**Table 2.** Current trials of hazelnut rootstocks in five countries.

| Rootstock | Country/Place/University | Reference |
|---|---|---|
| *C. colurna* seeds | Italy/Piedmont | [18] |
| | Italy/University of Perugia | [19] |
| | Italy/University Piacenza | [20] |
| *C. colurna* x *C. avellana* seedlings | Italy/University of Torino | [24,25] |
| 'Dundee' and 'Newberg' (open-pollinated *C. colurna* seedlings, with traits intermediate between *C. colurna* and *C. avellana*) | Spain/IRTA | [27] |
| | Italy/University of Torino | N. Valentini (pers. comm) |
| | Italy/University of Tuscia (Viterbo) | C. Silvestri (pers. comm) |
| | Romania/University of Craiova | I. Botu (pers. comm) |
| | Iran/University of Karaj | [32] |
| *C. avellana* cultivars | Italy/University of Torino | [25] |
| | Chile/Temuco (INIA) | [34] M. Ellena (pers. comm) |

## 3. Methods of Grafting Hazelnuts

Different grafting methods have been proposed for hazelnut [29,35–37], but not all of them have given good results. Improved techniques have increased grafting success, and with it, renewed interest in using rootstocks for this species. The authors of [38] made a major advancement in hazelnut propagation with the development of the "hot-callusing pipe", a simple and successful method for applying heat to the graft union alone, which is used outdoors or in an unheated building in winter. With this method, the warm temperatures necessary for good callusing in hazelnut (27 °C for 21–28 days) can be maintained at the graft union, while the rootstock and scion buds remain dormant under cool temperatures. With this device, which has over a 90% success rate, large-scale grafting can be an economically viable nursery practice. It is essential to have well-rooted rootstocks and to maintain moisture in the sawdust surrounding the roots. That graft union must also be completely sealed with overlapping wraps, rubber bands, plastic grafting bands, or masking tape. The cut end of the scion should be painted to prevent drying.

At IRTA–Mas Bové (northeastern Spain) chip budding is used in the nursery and gives good results. This type of grafting can be carried out in spring (April–May) or summer (late August–early September). Spring grafting requires that bud sticks be collected in winter (January), wrapped in plastic, and kept in refrigerated storage (4 °C). Fifteen days after the graft, the bud starts growing. For the summer graft, buds from the same season are needed. The rootstock and trees from which bud sticks are collected must be actively growing so that the bark will slip easily. In this case, the buds will remain dormant until the following spring.

Different grafting methods have been used in the trials mentioned in this manuscript, all of them with a high success rate. For trials using *C. colurna* seedlings as rootstocks, the English grafting method in April before the beginning of vegetation has been the most used [12,13,15,17]. When 'Dundee' and 'Newberg' were used as rootstocks, the "hot callusing pipe" system was used [24–27]. Finally, the same system was used when seedlings of *C. avellana* were used [31,34].

From different studies, it can be concluded that the grafting technique does not strongly influence the growth dynamics of hazelnut trees. All methods ensure normal

concrescence between the graft and rootstock, a successful graft union, and produce trees with normal development [36].

## 4. Current and Future Trends

The search for additional non-suckering rootstocks for commercial hazelnut orchards is continuing. Hybrids of *C. chinensis* x *C. avellana* have been mentioned [8,29] as potentially useful for developing vigorous and non-suckering rootstocks. Seedlings of *C. chinensis* are more vigorous and have more fibrous root systems than *C. colurna*—an issue that limits the use of *C. colurna* seedlings by the nursery trade. Adding to this, *C. chinensis* as a female parent hybridizes more readily with *C. avellana* and results in far fewer blank nuts than *C. colurna* [39].

In Chile, in 2014, the INIA research center in Temuco initiated a program to select clones of C. avellana with few or no suckers. Until now, two promising clones, 'SELFE J', and 'SELFE B' have been selected. These have been propagated "in vitro" and will be sent to Italy for evaluation in Europe. In southern Chile, 'Tonda di Giffoni' scions will be grafted on them, and several trials will be planted in 2022 in high-density orchards. They anticipate a yield increase from the grafted trees. In Chile, grafted trees with 'Tonda di Giffoni' scions and rootstocks 'Dundee' and Chilean selection 'Anders 5' are also being propagated for trials to be planted in 2022 at different agroecological sites. These activities are part of a project supported by Ferrero (Ellena, pers. comm). In the last 10 years, interest in hazelnut rootstocks has increased in both the research community and the producing sector. The Spanish nursery sector is producing grafted hazelnut trees, but not enough to satisfy the demand from growers. It has been reported that some Spanish nurseries have asked for 100,000 'Dundee' plants per year, and these are propagated "in vitro" in Spain. In Serbia, nurseries are annually producing approximately 50,000 seedlings of *C. colurna* for use as rootstocks and 30,000–50,000 grafted plants. In 2021, nurseries in Italy produced 100,000–150,000 trees grafted to *C. colurna* seedlings and also 30,000–40,000 bare-rooted *C. colurna* seedlings.

Currently, in Serbia, more than 1000 ha of hazelnut orchards have been planted of trees grafted to *C. colurna* seedlings (Roversi, pers. comm), and in Tarragona (northeastern Spain), young growers are pulling out commercial orchards more than 70–80 years old and replanting with grafted hazelnut trees. Currently, nearly 150 ha of commercial orchards of grafted trees have been established in this area. Nurseries and growers in Portugal, France, and Italy are also showing interest in planting grafted hazelnuts. This technology makes it easier to manage orchards and is transforming the hazelnut landscape in some areas, especially where traditional multi-stemmed shrubs were grown.

**Funding:** This research received no external funding.

**Institutional Review Board Statement:** Not applicable.

**Informed Consent Statement:** Not applicable.

**Acknowledgments:** The author gratefully acknowledges the valuable information about hazelnut rootstocks trials contributed by M. Botu, M. Ellena, A. Roversi, C. Silvestri, S. Tombesi, and N. Valentini, and the editing of the manuscript by S. A. Mehlenbacher.

**Conflicts of Interest:** The author declares no conflict of interest. The funders had no role in the writing of the manuscript.

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
