# Peer review of "Advances in Hazelnut (Corylus avellana L.) Rootstocks Worldwide"

_horticulturae, doi:10.3390/horticulturae7090267_

Round 1
Reviewer 1 Report
This work is interesting, especially for practitioners. It is not remarkably scientifically written and summarizes very little knowledge. Nevertheless, the simple goal set by the author was achieved. Paper is well prepared. I have marked a few minor comments for improvement in the PDF.

Author Response
Dear Reviewer,
Attahed you will finf the new version of my manuscript, following your comments.

Reviewer 2 Report
The ms 'Advances in hazelnut (Corylus avellana L.) rootstocks world-2 wide' is an exhaustive revision about not suckering rootstocks for hazelnuts.
The ms is well organized and well written with minors spelling mistakes.
Nonetheless it could be interesting to include a section related to compatibility/incompatibility of C. colurna L. with C. avellana
Author Response
Dear reviewer,
Thank you for your revision. You sugested me to include a section related to compatibility/incompatibility of C. colurna with C. avellana. This topic is develelopped already in the manuscript, but not in a special section. As you are the only reviewer (from 3 reviewers) who has proposed this new section, I consider that it is not necessary to include it.
Reviewer 3 Report
Interesting rewiew
Author Response
Thank you for revising my document